# Comparison of a Blood Self-Collection System with Routine Phlebotomy for SARS-CoV-2 Antibody Testing

**DOI:** 10.3390/diagnostics12081857

**Published:** 2022-07-31

**Authors:** Douglas Wixted, Coralei E. Neighbors, Carl F. Pieper, Angie Wu, Carla Kingsbury, Heidi Register, Elizabeth Petzold, L. Kristin Newby, Christopher W. Woods

**Affiliations:** 1Duke Clinical and Translational Science Institute, Duke University, Durham, NC 27701, USA; dougcarla.kingsbury@duke.edu (C.K.); kristin.newby@duke.edu (L.K.N.); 2Duke Global Health Institute, Duke University, Durham, NC 27710, USA; coralei.neighbors@duke.edu; 3Department of Biostatistics and Bioinformatics, Duke University Medical Center, Durham, NC 27710, USA; carl.pieper@duke.edu; 4Department of Clinical Research, Cytel Inc., Waltham, MA 02451, USA; angie.wu@cytel.com; 5Duke Clinical Research Institute, Duke University, Durham, NC 27715, USA; 6Duke Human Vaccine Institute, Duke University, Durham, NC 27710, USA; heidi.macht@duke.edu; 7Center for Applied Genomics and Precision Medicine, Duke University Medical Center, Durham, NC 27708, USA; elizabeth.petzold@duke.edu; 8Division of Cardiology, Department of Medicine, Duke University Medical Center, Durham, NC 27710, USA; 9Departments of Medicine and Pathology, Duke University Medical Center, Durham, NC 27710, USA; chris.woods@duke.edu

**Keywords:** COVID-19, coronavirus, SARS-CoV-2, Tasso-SST, capillary blood, self-collection, user acceptance, antibody testing, infectious disease

## Abstract

The Coronavirus Disease 2019 (COVID-19) pandemic forced researchers to reconsider in-person assessments due to transmission risk. We conducted a pilot study to evaluate the feasibility of using the Tasso-SST (Tasso, Inc, Seattle, Washington) device for blood self-collection for use in SARS-CoV-2 antibody testing in an ongoing COVID-19 prevalence and immunity research study. 100 participants were recruited between January and March 2021 from a previously identified sub-cohort of the Cabarrus County COVID-19 Prevalence and Immunity (C3PI) Study who were under-going bimonthly COVID-19 antibody testing. Participants were given a Tasso-SST kit and asked to self-collect blood during a scheduled visit where trained laboratory personnel performed routine phlebotomy. All participants completed an after-visit survey about their experience. Overall, 70.0% of participants were able to collect an adequate sample for testing using the device. Among those with an adequate sample, there was a high concordance in results between the Tasso-SST and phlebotomy blood collection methods (Cohen’s kappa coefficient = 0.88, Interclass correlation coefficient 0.98 [0.97, 0.99], *p* < 0.0001). The device received a high-level (90.0%) of acceptance among all participants. Overall, the Tasso-SST could prove to be a valuable tool for seroprevalence testing. However, future studies in larger, diverse populations over longer periods may provide a better understanding of device usability and acceptance among older participants and those with comorbidities in various use scenarios.

## 1. Introduction

The Coronavirus Disease 2019 (COVID-19) pandemic has forced healthcare institutions to reconsider clinical assessment and testing methods globally. Research study visits have an inherent risk for transmission of COVID-19 despite use of personal protective equipment (PPE), as evidenced by asymptomatic cases and outpatient nosocomial transmission and detection [1,2,3,4]. While many assessments can be accomplished via online tools, such as video conferencing capabilities or web-based surveys, some research studies may require specimen collection for laboratory testing. One way research studies can address this issue is the implementation of at-home specimen self-collection by participants. At-home specimen self-collection has been used by research studies in a variety of settings, such as testing for human immunodeficiency virus (HIV), sexually transmitted infection (STI), and rheumatoid arthritis (RA) [5,6,7,8]. In addition to reduced risk of transmission, benefits of implementing at-home self-collection in a research study include a reduced need for personnel and training and increased ability for large-scale testing [9]. In the context of COVID-19, one study found that people were willing to use at-home collection methods for a variety of sample types, regardless of demographics or the presence of COVID-19 symptoms [10]. Participants also reported that they would be more likely to participate in a research study with at-home self-collection compared with one that required clinic visits [10]. 

Since COVID-19 was discovered there have been many advances in diagnostics to detect COVID-19 infection, including antibody testing [11,12,13,14,15,16]. Antibody testing for severe acute respiratory syndrome coronavirus 2 (SARS-CoV-2), the causative agent of COVID-19, is used to detect antibodies (IgM, IgA, IgG, or total antibodies) from blood samples. Samples are typically collected via finger pricks or routine phlebotomy by a trained professional. A positive result indicates a possible previous COVID-19 infection. Seroconversion is dependent on the host and requires time; thus, antibody testing may not detect antibodies in participants currently experiencing a COVID-19 infection [17,18]. However, the use of wide-scale antibody testing is vital for epidemiological analyses exploring longitudinal antibody responses and immunity for COVID-19. To date, there are 85 serology and other adaptive immune response assays that have been granted a United States (US) Food and Drug Administration (FDA) emergency use authorization (EUA) for SARS-CoV-2 [19]. 

Currently, there are limited data regarding the reliability of remote sample collection for SARS-CoV-2 antibody testing. Many seroprevalence studies for COVID-19 have used dried blood spot cards collected remotely [20,21,22,23,24,25,26], but they can have a non-uniform blood distribution and uncertain collection volume [21,27]. One way to mitigate this issue of non-uniform blood distribution is the use of a volumetric sampling device, such as the Tasso-SST (Tasso, Inc, Seattle, Washington) device for capillary blood self-collection. The Tasso-SST is a single-use blood collection device intended for remote collection of capillary blood in individuals three months of age or older (https://www.tassoinc.com/tasso-sst, accessed 4 May 2022). The device is designed to collect 200 to 300 microliters of whole blood in 5 min or less and can store the sample without anticoagulation during transport to testing laboratories. This allows a receiving laboratory to perform serum separation and testing, without direct patient interaction from a trained phlebotomist or laboratory staff. Each device is individually shipped in a kit that includes all the necessary items, including instructions and return shipping materials. Currently, the Tasso-SST devices are available for investigational use only. 

Here, we describe our pilot study to evaluate the feasibility, user acceptance, and laboratory antibody test performance with the use of the Tasso-SST device for blood self-collection among adult research participants within the Cabarrus County COVID-19 Prevalence and Immunity (C3PI) Study [28].

## 2. Materials and Methods

### 2.1. Study Design

The Tasso Pilot Study was embedded within the conduct of participant visits for the C3PI Study. During a single, planned, in-person C3PI study visit, a blood sample for SARS-CoV-2 antibody testing was collected from consented participants via venipuncture performed by a trained phlebotomist. At the same visit, participants were given a Tasso-SST kit and asked to follow the kit directions to self-collect a capillary blood sample. An after-visit survey was sent via email to understand participants’ experience using the Tasso-SST kit. The study objectives were to (1) determine the failure rate of blood collection with the Tasso-SST kit, (2) compare the results of SARS-CoV-2 IgG antibody testing from serum collected using the Tasso-SST device with serum collected by routine phlebotomy, and (3) evaluate user acceptance of the Tasso-SST device. 

### 2.2. Participant Recruitment

Recruitment into the Tasso Pilot Study occurred from 15 January 2021 to 10 March 2021. All participants were recruited from the C3PI Study sub-cohort who underwent serial COVID-19 bimonthly antibody testing. The C3PI Study was a community COVID-19 surveillance study that enrolled 1410 individuals from the MURDOCK Study longitudinal cohort [29,30], and was conducted in North Carolina by Duke University with funding from the North Carolina Department of Health and Human Services (NCDHHS). The design, methods, and baseline characteristics of the C3PI Study and testing sub-cohort have been published previously [28].

A total of 281 participants in the C3PI testing sub-cohort with scheduled appointments and English as their preferred language were eligible to participate in the Tasso Pilot Study. Eligible participants received a recruitment email that included a participant-specific Duke REDCap link that provided a description of the study and the electronic consent form. Interested participants verified their identity by entering their first and last name, date of birth (DOB), and email address; all corresponding fields were then cross-referenced against the participant’s C3PI Study record to authenticate their identity. Database edit checks automatically alerted the study team to discrepancies that were reconciled via participant contact. Once participants verified their identity, they were prompted to complete the electronic informed consent form. 

Overall, 135 eligible individuals consented to participate in the Tasso Pilot Study. Due to the limited availability of Tasso-SST kits, only the first 100 consented individuals were enrolled. Forty participants completed study procedures during scheduled C3PI Study serology visits in January 2021; the remaining sixty participants completed study procedures during scheduled serology visits in March 2021. Participants were not offered compensation for their participation.

### 2.3. Study Visit Procedures

Following arrival and check-in, participants were escorted to an exam room, provided a Tasso-SST kit, and asked to read the step-by-step instructions included within the kit (Appendix A). The kits also included a QR code and a written web address linking to an instructional video on the Tasso website. Study staff observed blood collection but refrained from answering questions, instead referring participants to review the instructions as needed. Study staff entered the time the device was first adhered to the arm and the time the device was removed. After participants completed the Tasso-SST blood collection, trained study personnel collected blood via routine phlebotomy into a 3-mL red top tube according to the C3PI Study protocol. 

Tasso-SST samples were allowed to clot at room temperature for 30 min after collection. Tasso-SST tubes were then placed in standard vacutainers, as adapters, and centrifuged at 23 °C and 1500 g relative centrifugal force (RCF) for ten minutes to separate the serum as directed by the Tasso website (https://www.tassoinc.com/tassosst-lab-instructions, accessed on 3 December 2020). The 3-mL red top tubes collected from routine phlebotomy were allowed to clot at room temperature for forty-five minutes, then centrifuged at 4 °C at 3000 revolutions per minute (RPM) for 15 min per C3PI Study manual of procedures (MOP) [28]. Resultant serum was pipetted into 2 mL cryovials and stored at −80 °C until shipment. SARS-CoV-2 antibody testing was performed at the Immunology & Virology Quality Assessment Center (IQVAC), a Clinical Laboratory Improvement Amendments (CLIA) -approved laboratory within the Duke Human Vaccine Institute (DHVI) at Duke University in Durham, NC.

### 2.4. SARS-CoV-2 Antibody Testing

The IQVAC operator doing serology testing used the Abbott Alinity IgG nucleocapsid protein antibody assay with a serum minimum of 80 µL to account for void volume in the pipette [31]. All sample handling and equipment operation was directly followed per IQVAC standard operating procedure and manufacturer’s protocols. No deviations were implemented by the operator. The assay output was a sample/calibrator (S/C) ratio presented as an index result. Participants were considered to have SARS-CoV-2 antibodies (positive) if nucleocapsid IgG antibodies were detected at ≥1.40 index. The Abbott Alinity nucleocapsid IgG antibody assay has a specificity of 99.9% and a sensitivity of 100.0% for detecting IgG antibodies [32]. Participant samples were stored between 1–4 days prior to testing. To conduct testing, all samples were thawed at room temperature inside of a certified biosafety cabinet (BSC). Prior to processing, samples were transferred to sample vials using sterile pipette tips and a calibrated pipettor within a BSC. Pipette tips were changed between every sample transfer. Samples were then promptly transferred directly onto the instrument and processing was started by the operator. 

### 2.5. After-Visit Survey

Within 12 h of their study visit, participants received an automated survey email via REDCap. Email reminders were sent to non-responders, and phone calls were made as needed to ensure survey completion. The brief survey asked the following questions: Did you find the instructions for use of the Tasso-SST device easy to follow?Do you think you could use the Tasso-SST device to collect a blood sample at home and send the sample to the lab, if it was an option for the C3PI Study?Would you be willing to use the Tasso-SST device to collect a blood sample at home and send the sample to the lab, if it was an option for the C3PI Study?

Participants answered “yes”, “no”, or “don’t know/not sure” to each question. Answers of “no” or “don’t know/not sure” triggered the conditional question “Please describe why” to allow participants to provide details in a free-text field. At the end of the survey, participants could leave any additional comments in a free-text field. 

### 2.6. Statistical Analysis

Statistical analyses were conducted using Statistical Analysis Software (SAS) Version 9.4, SAS Institute Inc, Cary, North Carolina, USA. All baseline characteristics of the study population, nucleocapsid antibody testing results, and categorical survey responses were summarized using descriptive statistics (counts and percentages for discrete variables and means with standard deviations and medians with 25th and 75th percentiles for continuous variables). Differences in proportions were tested using the Fisher’s exact test, and medians were compared using the Student’s *t*-test. Both the Cohen’s kappa statistic [33] and McNemar’s tests were used to estimate the level of agreement and type of disagreement in antibody testing results between the Tasso-SST (experimental) and routine phlebotomy (“gold standard”) categorical results. Agreement between continuous values by the 2 assays were analyzed using the interclass correlation coefficient [34]. Additionally, a Bland–Altman [35] analysis was conducted to assess the agreement between routine phlebotomy and Tasso-SST results as continuous measures. 

## 3. Results

### 3.1. Study Population

Of 100 participants, a phlebotomy sample was available from all. Eighty-eight (88.0%) participants successfully collected a blood sample using the Tasso-SST device and 12 had no blood in the collection tube. Of the 88 Tasso-SST collections, serum could not be isolated from 5 (5.7%) due to low blood volume in the tube. The remaining 83 (94.3%) processed serum samples were sent to the IQVAC laboratory for SARS-CoV-2 antibody testing. Upon receipt and inspection at IQVAC, 13 (15.7%) samples did not have a sufficient quantity of serum for analysis, leaving 70 (70.0%) paired Tasso-SST and phlebotomy serum samples for nucleocapsid IgG antibody testing from the 100 enrolled participants (Figure 1). No samples had clear or apparent hemolysis during sample inspection and handling. The median time participants wore the Tasso-SST device was comparable between the primary analysis group (5.07 [5.03, 5.12]) minutes and those excluded for failures of sample collection, processing, or quantity sufficient for analysis (5.05 [5.03, 5.08]) minutes; *p* = 0.36.

### 3.2. Baseline Characteristics

Table 1 shows the baseline demographics and relevant characteristics of the primary analysis population, enrolled study population, and recruitment eligible population. Across the primary analysis, enrolled, and eligible populations, sex and highest education level were similar. For those who failed to collect a blood sample (*n* = 12), serum could not be isolated (*n* = 5), or were determined to have insufficient serum samples (*n* = 13), the median ages were 66.5 (59.0, 76.5), 73.0 (63.0, 75.0), and 62.0 (51.0, 73.0), respectively. Participants who were unable to collect blood successfully were significantly older than those who successfully collected a sample (66.5 [59.0, 76.5] vs. 58.0 [47.0, 70.5]; *p* = 0.03). Similarly, those excluded from the primary analysis population for any failure of sample collection, processing or volume were older than those included in the analysis (64.0 [53.0, 73.0] vs. 58.0 [45.0, 68.0]; *p* = 0.02). 

One hundred and fifteen (40.9%) of the recruitment eligible population had at least one predefined comorbidity that might affect ability to use the Tasso-SST device (e.g., arthritis) or the microvascular circulation and blood collection (e.g., diabetes mellitus). In comparison, 48 (48.0%) of the enrolled study population reported having at least one predefined comorbidity, as did 30 (42.9%) in the primary analysis population. Among the 30 participants with failed sample collection, 18 (60.0%) reported having one or more predetermined comorbidities (3 [16.7%] diabetes, 9 [50.0%] osteoarthritis, 1 [5.6%] rheumatoid arthritis, 1 [5.6%] diabetes and rheumatoid arthritis; 4 [22.2%] osteoarthritis and rheumatoid arthritis).

### 3.3. Comparison of SARS-CoV-2 IgG Antibody Testing Results 

Table 2 summarizes the dichotomous (positive, ≥1.40 index or negative, <1.40 index) nucleocapsid protein IgG antibody testing results for samples collected via the Tasso-SST device compared with samples collected via routine phlebotomy. Sixty-five of 70 (92.9%) phlebotomy samples were negative for nucleocapsid antibodies, and 5 (7.1%) were positive. There was one disagreement with the Tasso-SST results: an individual with a positive result in the routine phlebotomy sample who tested negative in the Tasso-SST sample (Cohen’s kappa coefficient = 0.88). Differences between the marginal frequencies were not significant (*p* = 0.31), indicating that the assay sensitivity using Tasso-SST specimen was equivalent to that using then routine phlebotomy sample.

Table 3 summarizes the continuous nucleocapsid protein IgG antibody results on paired samples. The antibody level by Tasso-SST sample collection was 0.02 (0.02, 0.07) and 0.04 (0.02, 0.09) by routine phlebotomy. The mean IgG nucleocapsid IgG antibody level by Tasso-SST collection was 0.43 (SD = 1.42) and 0.50 (SD = 1.48) by routine phlebotomy. The Interclass correlation coefficient between Tasso-SST and phlebotomy sample results was 0.98 (0.97, 0.99) (*p* < 0.0001). The continuous results for the paired samples with dichotomous disagreement were 0.11 and 2.42 for the Tasso-SST and phlebotomy samples, respectively. 

The Bland–Altman analysis plot is shown in Figure 2. There was very little bias in the differences between phlebotomy and Tasso-SST on average. As the average Tasso-SST and phlebotomy readings increased, there was a slight bias towards an increase in the phlebotomy and Tasso-SST difference, i.e., relative to the zero-change line, the regression line has a slight, positive bias. The bias between measurements (phlebotomy—Tasso-SST) was 0.72 ± 1.42 (limits of difference, 0.00 to 2.31). The high limit of difference is that between the paired samples with dichotomous disagreement.

### 3.4. User Acceptance of the Tasso-SST Device

Participant responses to the after-visit survey are summarized in Appendix B. Five participants (5.0%) did not think or were not sure whether the instructions for the Tasso-SST device were easy to follow. Of these participants, 4 (80.0%) successfully collected blood using the device. Nine (9.0%) participants did not think or were not sure they could use the device to collect blood samples at their house. Of these participants, 7 (77.8%) were unable to collect blood using the Tasso-SST device, and the remaining 2 (22.2%) had comfort/safety concerns. However, all nine participants felt the instructions were easy to use. Ten participants (10.0%) would not or were not sure whether they would be willing to use the device if it were an option. Eight (80.0%) of the 10 participants were among the 30 participants who could not collect a viable sample. When asked why, the 8 participants commented on their inability or difficulty to collect blood with the device, the convenience of visiting the research center, and not liking needles. However, most participants appreciated the device’s ease of use, convenience and time-saving ability compared with scheduled clinical visits, and minimal pain when using the device. Any specific comments made by those who responded “no” or “don’t know/not sure” to the conditional questions are shown in Appendix B.

## 4. Discussion

With the Tasso Pilot Study, we explored the performance and user acceptance of the Tasso-SST device, a capillary blood self-collection device, to obtain serum for use in SARS-CoV-2 IgG antibody testing. We found that an adequate sample for testing was collected using the Tasso-SST device 70.0% of the time. Among those with an adequate sample, there was a high concordance in antibody test results between the 2 blood collection methods, for both categorical (Cohen’s kappa coefficient = 0.88; *p* = 0.32) and continuous measures (Interclass correlation coefficient = 0.98; *p* < 0.0001) of antibody results. In addition, we also found a high-level (90.0%) of user acceptance among participants. 

The COVID-19 pandemic has led to the emergence of telemedicine as a substitute for in-person clinical and research study visits [36,37]. In this environment, access to a self-collection capillary blood device, such as the Tasso-SST device, could be helpful for widespread immunosurveillance studies of SARS-CoV-2 infections via nucleocapsid antibody tests, as well as other research serology-based testing for SARS-CoV-2 and other diseases, such as HIV or measles. Further, the ability to collect blood remotely could facilitate testing in other circumstances outside of a pandemic:, e.g., when travel distance, transportation barriers or disabilities limit access to routine phlebotomy. However, for self-collection to be helpful for clinical or research purposes, the performance of the self-collection method/device must be comparable with, if not better than, the current healthcare-assisted sample collection methods, including failure rate, user acceptance, and sample integrity. 

The high concordance between SARS-CoV-2 antibody assay results using capillary-derived serum collected by the Tasso-SST device and serum processed from blood collected by routine phlebotomy suggests that self-collection using this device may be a viable option. However, our observations on collection failure rates and user acceptance provide an important context for consideration of self-collection in future studies. Further, given our small analysis cohort and limited number of positive antibody tests, there is some inherent uncertainty in our concordance estimates.

We observed a sample collection failure rate with the Tasso-SST device of 30.0% in our study. Other studies that used the Tasso-SST device reported overall unsupervised failure rates of 6.6% and 0.3% [38,39]. The higher failure rate we observed could be due to our older population, in which manual dexterity or comorbidities could be more likely to affect use of the device. In our study, participants who did not collect an adequate sample with the Tasso-SST device were older than those who were successful with sample collection. The median age of our primary analysis group was 58.0 (45.0, 68.0) years. In other studies with higher self-collection success rates, participants were younger: 45 (range: 21–73) years and 20 (19, 21) years [38,39]. 

Beyond age, our population also had comorbidities that might affect the ability to use the Tasso-SST device. We found more participants with at least one prespecified comorbidity (diabetes, osteoarthritis, rheumatoid arthritis) among those with failed sample collection (60.0%) than in the primary analysis population (42.9%). Other studies that used the Tasso-SST device did not report such comorbidities [38,39]. 

Researchers will need to consider not only their target population, but also the assay they use for testing. Currently, 85 SARS-CoV-2 serology/antibody and other adaptive immune response tests have been granted an EUA by the FDA, each with its own serum volume requirements [19]. For this study, the Abbott Alinity nucleocapsid protein IgG antibody assay was used. The Abbott Alinity assay requires a minimum of 75 µL of serum; however, a minimum of 80 µL was used for this study to account for void volume in the pipette. One study using the Tasso-SST device reported a 6.6% failure rate with the CE-marked EuroImmun (Germany) anti-SARS-CoV-2 IgG kit, which has a 10 µL minimum serum requirement [38]. Another study used a combination of two assays [39]. The Abbott Architect SARS-CoV-2 IgG nucleocapsid assay was used as the primary assay, requiring a minimum volume of 75 µL. A Public Health England (PHE) in house receptor binding domain (RBD) assay was used when samples had an insufficient volume for the primary assay. A total of 211 (7.2%) samples could not be analyzed with the Abbott Architect SARS-CoV-2 IgG nucleocapsid assay due to insufficient volume; however, the study reported an overall failure rate of 0.3% (8/2913) due to use of the RBD assay when necessary. 

For a disease surveillance study, such as our C3PI Study, that requires serial testing over an extended period, a 30.0% failure rate would be problematic due to missed data points that could generate bias or diminish the external and internal validity of the study. In addition, an increase in effort, timelines, and operational costs could result when trying to correct the missed data points by sending out replacement kits or bringing participants with failed collections to the clinic for routine phlebotomy.

Use of the Tasso-SST device was generally well accepted in our study (90.0% reported they were willing to use it in the future), and participants reported that the device was generally easy to use (95.0% found instructions easy to follow and 91.0% thought they could use the device to collect a blood sample in the future). Of the participants who were not, or were not sure whether they were, willing to use the Tasso-SST device in the future, 80.0% (8/10) could not collect a viable sample. Participants who were not sure or were not willing to use the device in the future commented on their inability to use or difficulty with using the device, the preference of visiting the research center for a blood draw, and dislike of sticking themselves with needles. Most participants in our study appreciated the ease and minimal discomfort using the device as well as the convenience and time-savings of using the device compared with scheduled clinic visits. 

Our sample size was small, with few positive antibody results. The study was provided 100 Tasso-SST devices from the Henry Jackson Foundation, so a maximum of 100 participants could be enrolled. The sample collection failure rate of 30% further limited our sample size for comparative testing of antibody levels between the Tasso-SST device and phlebotomy to 70 participants. However, despite these limitations, the concordance between methods was high when compared, but should be confirmed in larger studies with a broader distribution of test results. Paired sampling and comparison of results were done at a single time for each enrolled participant. Sampling and comparison at two or more time points could have enabled a better understanding of failure rates, user acceptance, and estimates of concordance. In addition, our study processed all samples following their collection, including samples collected by the Tasso-SST device, which does not take into consideration the shipment prior to processing that is necessary for a home self-collection system and that could affect the stability of the sample and therefore result concordance. However, quantitative antibody levels have been found to be highly correlated between venous samples that were promptly delivered to a laboratory for testing and home-collected Tasso-SST capillary blood that were stressed with extreme shipping conditions [38]. Finally, although we collected user acceptance surveys for all participants, opinions may change over time with repeated use. 

Overall, the Tasso-SST device and similar self-collection methods could prove to be valuable tools for research testing that reduce potential exposures and provide convenience for participants. However, future studies in larger, diverse populations over longer periods of sampling may provide a better understanding of device usability and acceptance among older participants and those with comorbidities in various use scenarios.

## 5. Conclusions

The COVID-19 pandemic has forced researchers to reconsider research assessment and testing methods. In this setting, blood self-collection at-home could be useful in research. We found a high concordance in results for SARS-CoV-2 antibody testing using serum from capillary blood self-collection with the Tasso-SST device compared with serum prepared from blood obtained by routine phlebotomy by a trained professional. User acceptance of the Tasso-SST device was high, but the 30.0% failure rate for sample collection was a concern in our population.

## Figures and Tables

**Figure 1 diagnostics-12-01857-f001:**
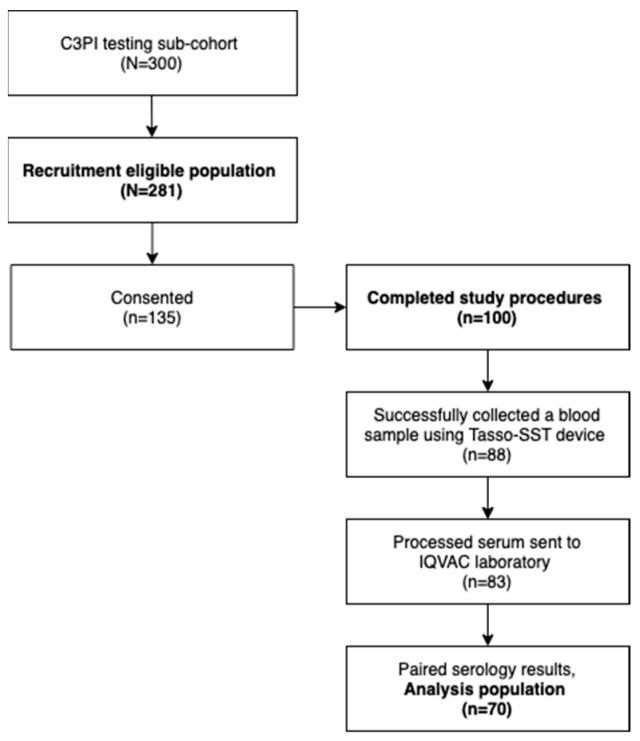
Study population.

**Figure 2 diagnostics-12-01857-f002:**
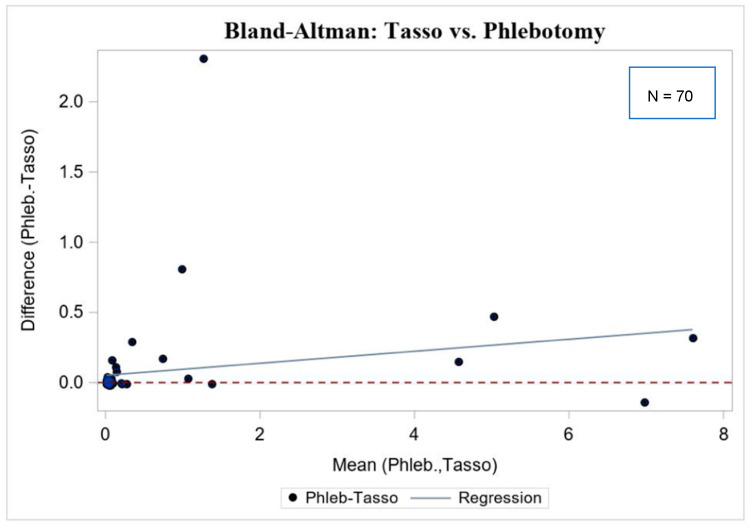
Bland Altman plot of continuous antibody testing results (*n* = 70).

**Table 1 diagnostics-12-01857-t001:** Baseline characteristics of the study analysis populations.

Variable ^1^	Primary Analysis Population (N = 70)	Enrolled Study Population (N = 100)	Recruitment Eligible Population (N = 281)
Age (in years)	58.0 (45.0, 68.0)	56.0 (49.0, 67.0)	57.0 (49.0, 68.0)
Male	27 (38.6)	40 (40.0)	108 (38.4)
Race			
White/Caucasian	57 (81.4)	85 (85.0)	226 (80.4)
Black or African American	8 (11.4)	10 (10.0)	46 (16.4)
American Indian or Alaska Native	1 (1.4)	1 (1.0)	1 (0.4)
Asian	0 (0.0)	0 (0.0)	0 (0.0)
Native Hawaiian or other Pacific Islander	0 (0.0)	0 (0.0)	0 (0.0)
Other	1 (1.4)	1 (1.0)	4 (1.4)
Multiple	3 (4.3)	3 (3.0)	4 (1.4)
Don’t know, not sure, prefer not to answer	0 (0.0)	0 (0.0)	0 (0.0)
Hispanic	7 (10.0)	7 (7.0)	18 (6.4)
Medical Comorbidities ^2^			
Osteoarthritis	23 (32.9)	36 (36.0)	89 (31.7)
Rheumatoid arthritis	9 (12.9)	15 (15.0)	27 (9.6)
Diabetes	7 (10.0)	11 (11.0)	34 (12.1)
COVID-19 Infection Status			
Positive study test	7 (10.0)	8 (8.0)	33 (11.7)
Days since positive study test	42.5 (37.0, 120.0)	43.0 (37.0, 168.0)	N/A
Highest Education Level			
Less than high school graduate	0 (0.0)	0 (0.0)	1 (0.4)
High school graduate/GED	4 (5.7)	7 (7.0)	15 (5.3)
Some college/associate’s degree	26 (37.1)	30 (30.0)	86 (30.6)
Bachelor’s degree	24 (34.3)	36 (36.0)	95 (33.8)
Master’s or higher professional degree	16 (22.9)	27 (27.0)	84 (29.9)

^1^ Categorical variables are represented by *n* (%) and continuous variables are represented by median (Q1, Q3). ^2^ All medical conditions are self-reported by participants as part of the MURDOCK Study enrollment questionnaire and/or annual follow-up form.

**Table 2 diagnostics-12-01857-t002:** Summary categorical IgG nucleocapsid antibody testing results.

		**Tasso-SST Device**		
		Negative	Positive	Total
**Phlebotomy**	Negative	65 (92.9%)	0 (0.0%)	65 (92.9%)
	Positive	1 (1.4%)	4 (5.7%)	5 (7.1%)
	Total	66 (94.3%)	4 (5.7%)	70 (100.0%)

**Table 3 diagnostics-12-01857-t003:** Summary continuous IgG nucleocapsid antibody testing results from primary analysis population.

Continuous Index	Samples Collected via Tasso-SST (*n* = 70)	Samples Collected via Routine Phlebotomy (*n* = 70)
Median (25th, 75th percentile)	0.02 (0.02, 0.07)	0.04 (0.02, 0.09)
Mean (SD)	0.43 (1.42)	0.50 (1.48)
Minimum, Maximum	0.00, 7.44	0.01, 7.76

## Data Availability

The data presented in this study are available on request from the corresponding author.

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
