# Peer review of "Comparison of a Blood Self-Collection System with Routine Phlebotomy for SARS-CoV-2 Antibody Testing"

_diagnostics, 2022, doi:10.3390/diagnostics12081857_

Round 1

Reviewer 1 Report

In the manuscript (Manuscript ID: diagnostics-1820771) entitle “Comparison of a Blood Self-Collection System with Routine Phlebotomy for SARS-CoV-2 Antibody Testing” by dr. Wixted and colleagues is a pilot study aimed in evaluating the feasibility of a remote blood self-collection device named Tasso-SST (Tasso, Inc, Seattle, Washington) for use in anti-SARS-CoV-2 antibody testing. A total of 100 participants were given a Tasso-SST kit and asked to self-collect blood during a scheduled visit where trained laboratory personnel performed routine phlebotomy. A total of 70 participants were involved in the primary analysis for comparing the two methods. Main results indicate the presence of a high concordance between the Tasso-SST and phlebotomy blood collection methods. The device received a 90% of acceptance among all participants. These findings indicate that Tasso-SST could be considered a possible tool for SARS-CoV-2 seroprevalence testing.

Overall, the work is interesting. The ms will increase our knowledge on remote self-collection devices for SARS-CoV-2 antibody testing. However, the ms can be improved in terms of data analysis and writing.  For instance, the introduction should be improved with more details on the state of the art on the remote sample collection methods/approaches for SARS-CoV-2 antibody testing. Contrariwise, the discussion is relatively well written and properly supports obtained results, highly informative, clear, and easy to follow. Several modifications/improvements should be applied in this work to making it suitable for publication (please see below)

I recommend a major revision

Major comments

The introduction is too poor and should be improved. Indeed, the majority of the introductive part is represented by the aim of the work and its experimental design. More information on remote blood self-collection devices and protocols for SARS-CoV-2 testing should be included. 

Besides Cohen’s kappa value, the Tasso-SST sensitivity and specificity and ROC curve analysis and relative AUCs should be computed on 70 paired serology results. These criteria are particularly specific for evaluating the clinical reliability of a test. Required data can all be taken from table 2

Minor observations

Line 20, SARS-CoV-2 should be “Severe acute respiratory syndrome coronavirus (SARS-CoV-2)” when mentioned for the first time. Please carefully revise the entire text for additional acronyms 

Line 43 “detection[1-4].” A space between words should eb included

If available, more supporting references should be included in 2.3, 2.4 and 2.6 methods sections

Lines 294-297 For completeness, these recent and detailed reviews on SARS-CoV-2 detection methods/protocols should be quoted in the introduction https://doi.org/10.3390/microorganisms10061193 , doi: 10.1007/s00604-022-05242-4 and PMID: 34896454

How many samples have been included in figure 2? They look less than 70 samples

Subhead titles should be removed from the discussion for a better reading of the text. It would be helpful for the reader

Line 302 and 325 the text of these lines is difficult to read

Line 321 I discourage starting a sentence with a number

Author Response

We appreciate the careful review or our manuscript. We have responded to each of the reviewer’s comments below. The original comments of the reviewer are in bold font, followed by our responses.

REVIEWER 1

Date of review   11 Jul 2022 21:07:33

Summary

In the manuscript (Manuscript ID: diagnostics-1820771) entitle “Comparison of a Blood Self-Collection System with Routine Phlebotomy for SARS-CoV-2 Antibody Testing” by dr. Wixted and colleagues is a pilot study aimed in evaluating the feasibility of a remote blood self-collection device named Tasso-SST (Tasso, Inc, Seattle, Washington) for use in anti-SARS-CoV-2 antibody testing. A total of 100 participants were given a Tasso-SST kit and asked to self-collect blood during a scheduled visit where trained laboratory personnel performed routine phlebotomy. A total of 70 participants were involved in the primary analysis for comparing the two methods. Main results indicate the presence of a high concordance between the Tasso-SST and phlebotomy blood collection methods. The device received a 90% of acceptance among all participants. These findings indicate that Tasso-SST could be considered a possible tool for SARS-CoV-2 seroprevalence testing.

Overall, the work is interesting. The ms will increase our knowledge on remote self-collection devices for SARS-CoV-2 antibody testing. However, the ms can be improved in terms of data analysis and writing.  For instance, the introduction should be improved with more details on the state of the art on the remote sample collection methods/approaches for SARS-CoV-2 antibody testing. Contrariwise, the discussion is relatively well written and properly supports obtained results, highly informative, clear, and easy to follow. Several modifications/improvements should be applied in this work to making it suitable for publication (please see below)

I recommend a major revision

Author response: Thank you for the detailed review and thoughtful feedback. We have addressed all comments and included responses to each below.

Major comments

The introduction is too poor and should be improved. Indeed, the majority of the introductive part is represented by the aim of the work and its experimental design. More information on remote blood self-collection devices and protocols for SARS-CoV-2 testing should be included. 

Author response: We have expanded the introduction to incorporate more information on remote blood self-collection devices and protocols for SARS-CoV-2 antibody testing as suggested. We believe that the revised introduction is improved as recommended, while preserving its structure to introduce the aim of the research and experimental design.

Besides Cohen’s kappa value, the Tasso-SST sensitivity and specificity and ROC curve analysis and relative AUCs should be computed on 70 paired serology results. These criteria are particularly specific for evaluating the clinical reliability of a test. Required data can all be taken from table 2

Author response: Our primary aim was to understand whether the results of the serology testing were comparable between POC capillary blood collection via the Tasso-SST device and blood collected via phlebotomy. Although routine phlebotomy is the established method for collecting blood for serology testing, it is not a ‘gold standard’ or ‘truth’ against which we can compare the Tasso-SST. Since there is no “truth”, rather there are two distinct and fallible completion methods for comparison, statistics like Sensitivity, Specificity and AUC which require ‘truth’ are not appropriate.

Minor observations

Line 20, SARS-CoV-2 should be “Severe acute respiratory syndrome coronavirus (SARS-CoV-2)” when mentioned for the first time. Please carefully revise the entire text for additional acronyms 

Duke response: We have made this edit and searched on all acronyms to ensure they are defined at first use.

Line 43 “detection[1-4].” A space between words should eb included

Author response: A space was included in revised manuscript.

If available, more supporting references should be included in 2.3, 2.4 and 2.6 methods sections

Author response: We have reviewed our manuscript and revisited our literature review in order to incorporate more supporting references in our introduction and throughout our methods sections as suggested.

Lines 294-297 For completeness, these recent and detailed reviews on SARS-CoV-2 detection methods/protocols should be quoted in the introduction https://doi.org/10.3390/microorganisms10061193 , doi: 10.1007/s00604-022-05242-4 and PMID: 34896454

Author response: Thank you for sharing these reviews.  We have incorporated them into our introduction and added them to our references.

How many samples have been included in figure 2? They look less than 70 samples

Author response: Figure 2 includes 70 data points; however, most are at or around the 0, 0 location and create a cluster in that region. We have added a label to the figure to clarify n=70 data points. 

Subhead titles should be removed from the discussion for a better reading of the text. It would be helpful for the reader

Author response: We have removed these subheadings in the revised manuscript as suggested by the reviewer.

Line 302 and 325 the text of these lines is difficult to read

Author response: Thank you for this feedback. We have attempted to improve readability here and wherever possible based on the reviewer’s suggestions.

Line 321 I discourage starting a sentence with a number

Author response: We have addressed this in our revised manuscript.

Reviewer 2 Report

The present manuscript represents a very well executed comparison of diagnostic procedures. Two aspects of the evaluated device, which are “ (1) general practical applicability” and “(2) use in SARS-CoV-2 antibody testing” were thoroughly investigated and results were presented in a very structured, clear way. Negative features of the evaluated device derived from the results (e.g. 30% failure rate in this specific study population) were sufficiently discussed. Statistics performed appears rock solid, to the extent that I can judge (as I am not a statistician by training). The use of scientific English language is immaculate. No typos were apparent, which made it a pleasure to read this manuscript (as in my experience the majority of to-be-reviewed manuscripts in international journals appear sloppy in spelling, phrasing and formatting). Scientific value is given to a certain extent, as the evaluation of diagnostics procedures provides the basis for clinical studies. Novelty of the manuscript is inherent to the specific application of a certain device on a patient cohort with certainly unique characteristics. Sample size is modest, but sufficient for the presented conclusions. Congratulations to the authors on this work!

Author Response

We appreciate the careful review or our manuscript. We have responded to each of the reviewer’s comments below. The original comments of the reviewer are in bold font, followed by our responses.

REVIEWER 2

Date of review   14 Jul 2022 16:49:25

Summary

The present manuscript represents a very well executed comparison of diagnostic procedures. Two aspects of the evaluated device, which are “ (1) general practical applicability” and “(2) use in SARS-CoV-2 antibody testing” were thoroughly investigated and results were presented in a very structured, clear way. Negative features of the evaluated device derived from the results (e.g. 30% failure rate in this specific study population) were sufficiently discussed. Statistics performed appears rock solid, to the extent that I can judge (as I am not a statistician by training). The use of scientific English language is immaculate. No typos were apparent, which made it a pleasure to read this manuscript (as in my experience the majority of to-be-reviewed manuscripts in international journals appear sloppy in spelling, phrasing and formatting). Scientific value is given to a certain extent, as the evaluation of diagnostics procedures provides the basis for clinical studies. Novelty of the manuscript is inherent to the specific application of a certain device on a patient cohort with certainly unique characteristics. Sample size is modest, but sufficient for the presented conclusions. Congratulations to the authors on this work!

Author response: Thank you for the detailed review and thoughtful feedback. The authors sincerely appreciate your congratulations and support of the manuscript.

Round 2

Reviewer 1 Report

The manuscript can be accepted in the present form